# The Pyrogeography of Methane Emissions from Seasonal Mosaic Burning Regimes in a West African Landscape

Paul Laris [1,*], Moussa Koné [2], Fadiala Dembélé [3], Christine M. Rodrigue [1], Lilian Yang [1], Rebecca Jacobs [1], Quincy Laris [4] and Facourou Camara [5]

1  Geography Department, California State University Long Beach, 1250 Bellflower Blvd, Long Beach, CA 90840, USA
2  Institut de Géographie Tropicale (IGT), UFR-SHS, Université FHB de Cocody-Abidjan, Cocody-Abidjan 21, Abidjan BP 5329, Côte d'Ivoire
3  Institut Polytechnique Rural de Formation et de Recherche Appliquée de Katibougou, Koulikoro BP 06, Mali
4  Department of Civil and Environmental Engineering, University of California, Berkeley, CA 94720, USA
5  Independent Researcher, Tabou Village, Siby BP 48, Mali
*  Correspondence: paul.laris@csulb.edu

**Abstract:** People have set fire to the savannas of West Africa for millennia, creating a pyrogeography. Fires render the landscape useful for many productive activities, but there is also a long history of efforts to regulate indigenous burning practices. Today, savanna fires are under scrutiny because they contribute to greenhouse gas emissions, especially methane. Policy efforts aimed at reducing emissions by shifting fire regimes earlier are untested. Most emissions estimates contain high levels of uncertainty because they are based on generalizations of diverse landscapes burned by complex fire regimes. To examine the importance of seasonality and other factors on methane emissions, we used an approach grounded in the practices of people who set fires. We conducted 107 experimental fires, collecting data for methane emissions and a suite of environmental variables. We sampled emissions using a portable gas analyzer, recording values for CO, $CO_2$, and $CH_4$. The fires were set both as head and backfires for three fire periods—the early, middle, and late dry season. We also set fires randomly to test whether the emissions differed from those set according to traditional practices. We found that methane emission factors and densities did not increase over the dry season but rather peaked mid-season due to higher winds and fuel moisture as well as green leaves on small trees. The findings demonstrate the complexity of emissions from fires and cast doubt on efforts to reduce emissions based on simplified characterizations of fire regimes and landscapes.

**Keywords:** savanna fires; methane; emission factors; combustion efficiency; West Africa

## 1. Introduction

People have been setting fire to the savannas of West Africa for millennia. The fire regimes are strategic, not random, as fires are typically set in a spatiotemporal pattern, creating a unique pyrogeography. Fires are set for a plethora of reasons and generally render the savanna landscape more useful to the local population for a variety of productive activities. Of course, some fires occur accidentally, some are malicious, and even purposefully set fires can burn outside of their intended areas with potentially negative consequences. Whether a particular fire is beneficial or detrimental depends upon numerous factors, such as intended land use, and a host of variables, including the timing and location of the fire. There has been a long history of efforts to regulate and control indigenous practices of setting fires. History has shown us that these efforts often fail and sometimes have disastrous long-term consequences.

Today, it is known that savanna fires contribute significantly to greenhouse gas (GHG) emissions, especially methane [1]. Yet, while it is recognized that these fires play an important role in the global methane cycle, there are few accurate estimates of emissions

and no recent ones from West Africa, one of the continent's most active fire regions. Most estimates of GHG emissions contain high levels of uncertainty [2,3] (in part because they are based on generalizations of diverse landscapes that are burned by complex fire regimes, creating a specific pyrogeography) [4].

Savanna complexity is both spatial and temporal, and it is a function of human and natural factors. While the spatial complexity of savanna landscapes is well recognized in the savanna literature (e.g., [5,6]), their temporal complexity has received less attention. Research has documented, for example, that the ratio of trees to grasses can vary dramatically across savanna landscapes depending upon a suite of factors, including edaphic conditions and disturbance regimes [6–8], but the impacts of these spatial patterns on the GHG emissions of fires have been less studied. Far less is known about the relationship between temporal complexity and fire emissions, although indigenous fire users clearly take advantage of temporal changes in vegetation to manage landscape fires. Savannas have been most studied from the perspective that dry season drought causes a gradual desiccation of vegetation, which results in seasonal changes in fire hazard, intensity, and severity (e.g., [9–11]). Depending on the fire regime, gradual drying can also result in changes in the size and contiguity of burned patches (e.g., [12]) as well as the type and state of the fuels burned. Unfortunately, research on seasonal changes in fires and their emissions is limited and has been hampered by a persistent underlying bias involving the overreliance on crude (binary) measures of seasonality based on early/late season fires (e.g., [13]). The use of the fire binary hampers the development of accurate emissions models as it fails to adequately capture the many temporal shifts in the state of savanna vegetation and fires over time, potentially resulting in biases in research as well as policy recommendations [14,15]. To give but one example, tree leaf drop, which occurs in the middle dry season (MDS), alters fuel structure and fuel loads, but its effect on combustion and emissions has been little studied.

Anthropogenic fires dominate Africa. In West Africa in particular, the anthropogenic nature of fire regimes removes much of the inter-annual variation common elsewhere as they are a regular, predictable feature of the landscape often with a distinct spatiotemporal pattern (e.g., [16]. Policymakers have long criticized anthropogenic fires for reducing tree cover and contributing to land degradation or deforestation. As a result, colonial and independent governments periodically tried to eradicate landscape burning. These efforts had little success and sometimes produced disastrous outcomes, including a rise in larger and more catastrophic fires [17], mirroring a broader global problem [18]. Efforts at regulation have waxed and waned with the political context, drought cycles, and periods of international concern. Rural residents frequently perceive government fire restrictions as an imposition on their way of life, and enforcement has led to animosity towards government agents and, more recently, the collapse of government-supported fire management efforts (e.g., [19]).

Today there is a new and growing effort to regulate anthropogenic fire regimes in savannas, driven by an international effort to reduce GHG emissions [20,21]. The basic premise is that improved fire management, especially in tropical Africa, can reduce carbon emissions and even provide much needed financial support to local economies ([21] teal). While the incentive system for this new model is novel—based on a carrot (money for carbon offsets) rather than a stick (fines for setting fires) approach—the guiding philosophy is the same: the burning practices of local land managers cause environmental damage and need to be regulated and improved. In the colonial and early postcolonial eras, the goal of policies was to reduce fires—especially late dry season (LDS) fires—to lessen tree damage [17,22]. The efforts made today are similar as they also aim to reduce LDS fires based on the notion that fires set in the early dry season (EDS) burn less area more patchily, and thus consume less biomass, which theoretically results in lower methane emissions (Lipsett [20]). The irony of the new carbon offset efforts is that early burning is, and has long been, a staple strategy of fire use in many African countries and it remains the dominant form of burning in West Africa [11,16,23,24].

In sum, while it is theoretically possible that a shift to earlier burning in some areas will result in a reduction in methane emissions, the science is insufficiently developed for sweeping policy suggestions [20,21] as several fire scholars have noted [15,25]. Specifically, while there have been a few recent efforts to incorporate some effects of temporal complexity into models of emissions estimates, few studies have sufficiently dealt with the factors affecting the changes in seasonal emissions from fires and little research has attempted to integrate both spatial and temporal aspects of savanna complexity into emissions estimates and models. Moreover, there have been very few efforts to link actual human burning practices to the fire regimes they produce and their methane emissions. The objective of this study is to provide a case study of how complexity—of both burning practices and savanna landscapes—can influence fire characteristics and ultimately, methane emissions.

To understand the effects of such human ecological complexity on fires and the methane emissions produced, we designed our experiments to set fires according to the well-established seasonal mosaic burning regime of local inhabitants [16,26]. Specifically, we set over 100 fires to specific patches of savanna vegetation mostly in accordance to local practices in order to estimate their emissions. For comparison, we also set some fires randomly to different patches of savanna. To improve upon the binary fire model of seasonality, we divided our fire experiments into three distinct seasons: the early, middle, and late dry season (EDS, MDS, and LDS). We note that a major problem with the fire binary is that there is no recognized measure for dividing early- from late-season fires. As such, we attempted to create distinct burning seasons based on both knowledge of local burning practices and logic [26,27] as well as an examination of several long-term fire databases. Nonetheless, we recognize that the distinction of specific fire seasons remains somewhat arbitrary, as one such goal of this study is to critically examine how key factors that shape fire regimes—and their emissions—vary by season in hopes of contributing to acceptable metrics for distinguishing between fire seasons. Our overall objective was to understand the role of seasonality on methane ($CH_4$) emission factors (EF), a measure of the mass of $CH_4$ emitted per mass of fuel burned (usually depicted in g/kg), and emission density (ED), the mass of $CH_4$ emitted per unit area burned (usually depicted in $g/m^2$). Specifically, this required determining how key variables thought to affect emissions vary by season, including ambient air conditions, wind, and fuel moisture, type, and load. These variables in turn affect key burn factors, including combustion efficiency, completeness, and fire intensity.

### 1.1. Methane Emissions and Savanna Spatiotemporal Complexity

To determine how spatiotemporal complexity and seasonality affect methane gas emissions from savannas, we begin by modifying the standard equation used to quantify the gas species emitted from vegetation fires based on the biomass burning emissions model of the Intergovernmental Panel on Climate Change (IPCC) ([28]: 49, [29]: A2.13):

$$\text{Emission (tons)} = \text{Burned Area (ha)} \times \text{Fuel (tons/ha)} \times \text{Completeness (\%)} \times \text{Emission Factor (g·kg}^{-1}) \times 10^{-3} \quad (1)$$

Here, Emission is the quantity of gas or aerosol flux in tons; Burned Area is the total area burnt in hectares; Fuel is the total load of burnt biomass in tons per hectare; Completeness is the fraction of fuel pyrolyzed fire expressed as a percentage; and Emission Factor of a gas is the amount of gas generated when one kilogram of fuel is burnt. To understand how seasonality can affect emissions, we examine each factor and revise this formula to include seasonally specific values for the area, fuel load, combustion completeness, and emission factor and add the variable burned area completeness (BAC), which we define as the fraction of the surface area affected by the fire expressed as a percentage.

Beginning at the broadest temporal scale, the movement of the Intertropical Convergence Zone (ITCZ), which controls broad shifts in the weather, including precipitation, humidity, and wind patterns, drives seasonality in West African savannas. Critically, as the dry season progresses, soil moisture declines as does vegetation (fuel) moisture. This

process is spatially uneven due to distinctions in both edaphic conditions and vegetation characteristics (e.g., perennial vs. annual grasses). Typically, patches of shorter annual grasses (often on poor, shallow, lateritic soils) dry first, with taller annuals on better soils drying later followed by perennials on deeper soils or in flood plains. The gradual desiccation over space and time results in a patchy mosaic of vegetation—fuels—at differing levels of dryness (fuel moisture content). This heterogeneity—patchiness is ever shifting during the dry season as areas with higher soil moisture increasingly dry over time. The predominate fire regime in the region is based upon the logic of local burning practices which progressively burns the drying vegetation, creating a regularized regime of seasonal mosaic fires (Figure 1) [16,17,26]. Finally, at the end of the dry season, small fires are set to prepare fields for agriculture.

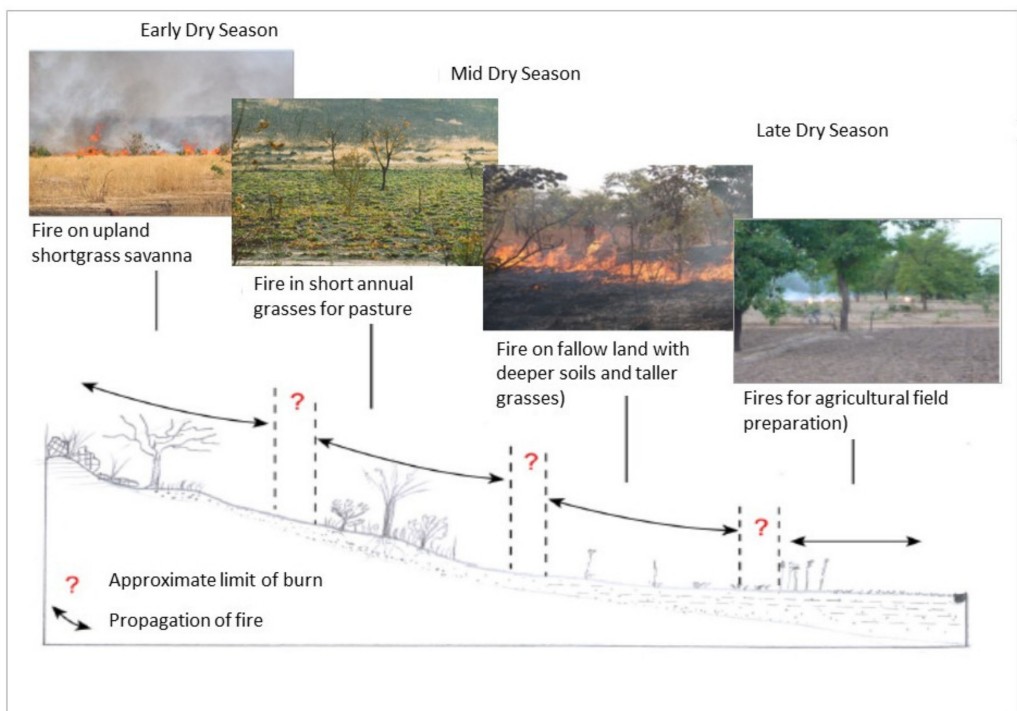

**Figure 1.** Schematic of spatiotemporal regime of landscape burning based on vegetation type and seasonality for West Africa (modified from [24]).

In addition to the critical changes in fuel moisture (FM), other key factors also shift over the course of the dry season, which typically runs from November through May in West Africa. In general, humidity falls over the course of the dry season (with the minor exception of occasional light, so-called mango rains), soil moisture falls (albeit spatially unevenly), and temperature initially falls in the Harmattan period (December and January) and then increases. Winds increase from November to January as the Harmattan blows out of the northeast before peaking and declining later in the dry season during March. Leaf fall also commences in this period, typically beginning in December [30].

As the description above illustrates, there is no simple temporal pattern to justify a binary view of savanna dry seasons. While it is true that the overarching pattern is one of a progressively drying landscape, the pattern is spatially and temporally heterogeneous and the pattern of heterogeneity matters. When combined, the basic shifts in climate and vegetation cause numerous effects on the factors that determine fire properties, including fuel type and conditions, fuel load, fuel combustion, combustion completeness, and patchiness. The precise impacts of these are little studied. There is evidence to suggest that the "winter" period of late December–January is unique, with lower temperatures and higher winds than the earlier and later periods of the dry season. Winter leaf drop is also a critical point in the dry season with key implications for combustion and emissions (see

below). It is also important to note that the harvest season in West Africa typically begins in late November and runs through December. People are involved in harvesting crops and have less time to manage fires; as such, setting fire to dry grasses interspersed with fields of unharvested, very flammable crops of cotton, corn, and millet is extremely risky and generally discouraged or banned. Indeed, fine resolution analyses of the temporal regimes of fires often document a "dip" in the number of fires and the area burned during this period [17]. In general, vegetation is gradually desiccated over the course of the dry season; the progression is temporally and spatially uneven due to the differences in vegetation and soil moisture. This creates a seasonal mosaic pattern, which people use to create a patch-mosaic burn regime especially when burning commences early in the dry season, as evidence has suggested.

Based on the general description of seasonal changes in the savanna described above, we postulate the following three seasonal impacts on fire emissions:

1. Vegetation desiccation (fuel drying) generally causes an increase in combustion efficiency as drier fuels combust more completely, which *theoretically results in a drop in the EFs of CO and CH$_4$ since they are products of incomplete combustion*.

2. As fuel drying progresses further, there is an increase in fire intensity (a measure of the energy emitted by a fire, which affects the flame and scorch height), as well as fire severity (the amount of vegetation affected by a fire), which *theoretically increases the amount of fuel consumed and increases the emission density*.

3. As grasses (fuels) become more uniformly dry as the season progresses, there is less patchiness in terms of fuel moisture, typically resulting in an increase in burn size and completeness (both the CC and BAC) at the plot and landscape scale, *theoretically increasing the emission density* (Figure 2).

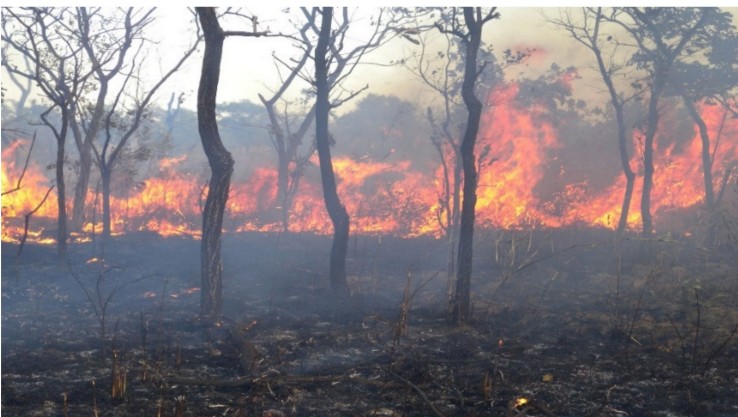

**Figure 2.** A mid-season head fire in a wooded savanna in Faradiélé, Mali. Note the flame height is relatively high (>2 m), indicating a high-intensity, wind-driven head fire. Smoke indicates a lower combustion efficiency even though burned area and combustion completeness appear very high (Photograph by P. Laris).

Based on these three postulates, we theorize that counteracting forces operate to determine methane emissions. We predict that the CH$_4$ EF will decline over the course of the dry season due to the drying of fuels, while the BAC and CC will increase as the landscape becomes more uniformly dry, having opposing influences on the CH$_4$ density (Figure 3). There are many caveats, adding further levels of complexity. First, as noted, winds change over the dry season, increasing mid-season for West Africa. Increasing winds can enhance fire spread rates and cause an increase in fire intensity (especially for head fires). Second, leaf fall, which begins mid-season, contributes to an increase in fuel load, but also changes the fuel structure. Leaf litter increases fuel connectivity, but decreases fuel aeration as well as fuel location. These changes have little-known impacts on the CH$_4$ EF, CC, and BAC. Finally but importantly, human practices have numerous effects on burning

and combustion. People tend to set backfires rather than head fires, which burn more slowly, with less intensity and lower flame height, but with high completeness. People also tend to set fires in the late afternoon after the winds and temperature fall while the humidity rises, decreasing the fire intensity and severity while increasing the patchiness [26,27]. Finally, we emphasize that this model does not account for the *actual* burning practices of people, which will ultimately determine the conditions under which a fire is set—especially the weather and fuel moisture conditions. People set fires in accordance with the drying rates of different grass species on different patches of savanna landscapes and tend to set fires late in the day when the winds and temperature are dropping. As such, burning regimes are not random, nor haphazard, but rather are quite systematic with specific patches of vegetation burning at particular times in the dry season on an annual basis [24,27]. As such, the point at which burning should commence or halt in order to reduce methane emissions must be determined empirically through studies of shifting emission factors and combustion levels, as well as spatiotemporal analyses of fires. Our study explicitly considers this while also conducting some "random" burns in the middle dry season for comparative purposes.

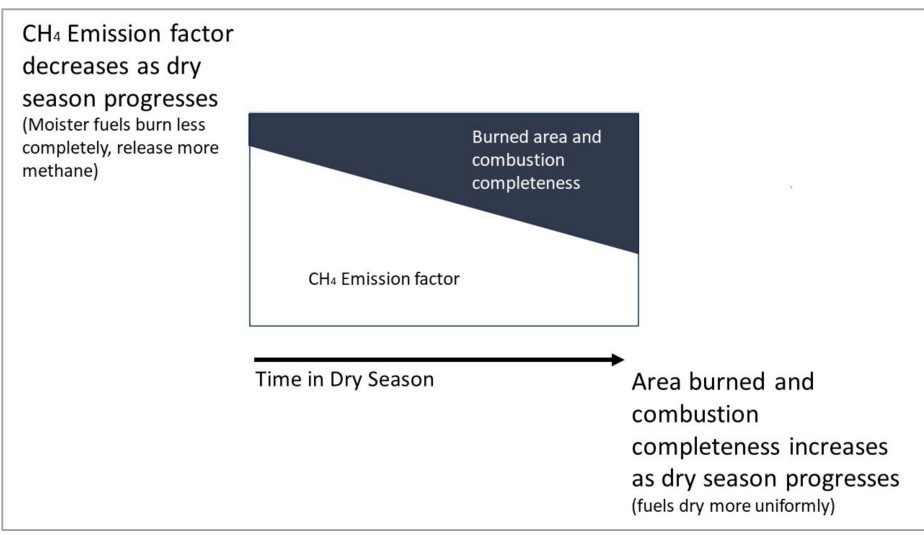

**Figure 3.** Theoretical competing determinants of methane emission density in a savanna landscape by season.

### 1.2. Seasonal Emission Equations

To account for the seasonality effects described above, we revise the IPCC general emissions formula to include seasonally specific values for area, fuel load, combustion completeness, and emission factors and add the variable BAC to account for the fraction of the surface area affected by the fire. We suggest the following revision for determining emissions *by fire season* in savannas ($E_s$):

$$\text{Emission}_s \text{ (tons)} = BA_s \text{ (ha)} \times FL_s \text{ (tons/ha)} \times CC_s \text{ (\%)} \times EF_{xs} \times BAC_s \text{ (\%)} \qquad (2)$$

Here, $BA_s$ is the burned area, $FL_s$ is the fuel load, $CC_s$ is the combustion completeness, $EF_{xs}$ is the emission factor of species $x$, and the $BAC_s$ is the burn area completeness by each season (E-M-L). We note that some estimates include the additional factor of landscape scale patchiness, which adjusts for the fraction of unburned area not captured by the image analysis. We assume this to be low in the 30 m Landsat data, but potentially higher in the oft-used MODIS data due to the scale of burned area mapping [31]. Note that we did not determine the total BA for this work, but have done so previously using Landsat data [27]. As such, we present our results in terms of emission density (emissions per meter squared) according to the season of the fire.

Finally, to distinguish the many factors that influence emissions from fires, it is helpful to depict the variables by scale (Figure 4). At the scale of gas emissions in smoke, a key term is the combustion efficiency (CE), which is defined as the ratio of $CO_2$ to all the carbon-based gases. The CE is most often represented as the modified combustion efficiency (MCE)—the ratio of carbon dioxide to carbon monoxide plus carbon dioxide ($CO_2/(CO + CO_2)$). The MCE is a widely used metric to distinguish flaming (MCE > 0.90) from smoldering or smoky combustion (Figure 4a). At the level of fuel combustion, the CC is the fraction of the fuel by weight that is pyrolyzed during a fire. The CC is effectively the fuel weight minus the moisture content, ash, and unburned biomass that remain after a fire (2b). Finally, at the level of the burned patch, the BAC is a visual measure of the percentage of the surface area affected by a fire. The BAC is scale-dependent and can be measured at either the landscape or the plot scale. We measured the BAC at the plot scale, defining it as the visual fraction of the 100 (10 * 10) m$^2$ plot affected by a fire (2c). At the landscape scale, the BP is often a function of the technology used to map a fire. For example, when satellite imagery is used, the BP can be thought of as a measure of the landscape patchiness. It can be defined either as the fraction of a "pixel" burned that is affected by a fire [32] or it can be characterized according to landscape metrics [12].

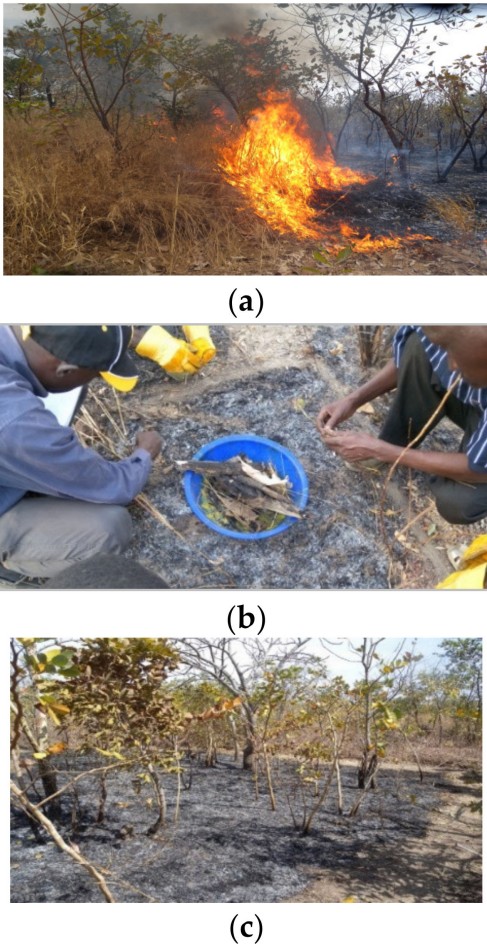

**(a)**

**(b)**

**(c)**

**Figure 4.** Scales of fire emission analysis. (**a**) The scale of combustion efficiency (CE) of emissions. Inefficient combustion results in a smoky fire with higher levels of CO and lower levels of $CO_2$, leading to a lower MCE. (**b**) Scale of combustion completeness (CC) (ash and residuals). Incomplete combustion results in lower CC. (**c**) Scale of plot, uniformity of burn completeness, or BAC. The plot shown has very high BAC (nearly 100% for fine fuels). It also has low scorch height (<1 m) typical of a low-intensity, slow-burning backfire.

## 2. Methods

To determine the factors that most affect fire emissions of methane gas, we conducted 107 experimental burns using a field-based method to measure key factors. Vegetation plots, time of day of fire, and season (early, middle, or late) were selected based on local burning practices. We collected data for savanna type, grass type, biomass composition and amount consumed, scorch height, speed of fire front, fire type, and ambient air conditions for two mesic savanna sites in Mali. We used regression analysis to determine the key factors affecting methane EF and density values. Our research was conducted in two working landscapes located in the Sudanian savanna of southern Mali (Figure 5). We chose areas with precipitation over 900 mm because they burn frequently and are known to be fire-determined landscapes. The climate can be divided into two seasons: a wet period from approximately June through October and a dry season from November through May. The dry season can be further subdivided into a cool dry period from approximately November through February and a hot dry period from March through May. This distinction is important for fires, because weather in the cool period is dominated by the Harmattan wind, which is dry, desiccates vegetation, and creates unique fire weather. The winds generally wane in the later dry season while temperatures rise. The mean annual rainfall is 991.2 mm for Bamako and 1176.8 mm for Bougouni (urban centers in each region) [33]. The fire season follows the rains and typically runs from November through April, with the bulk of the burning occurring in late December and early January.

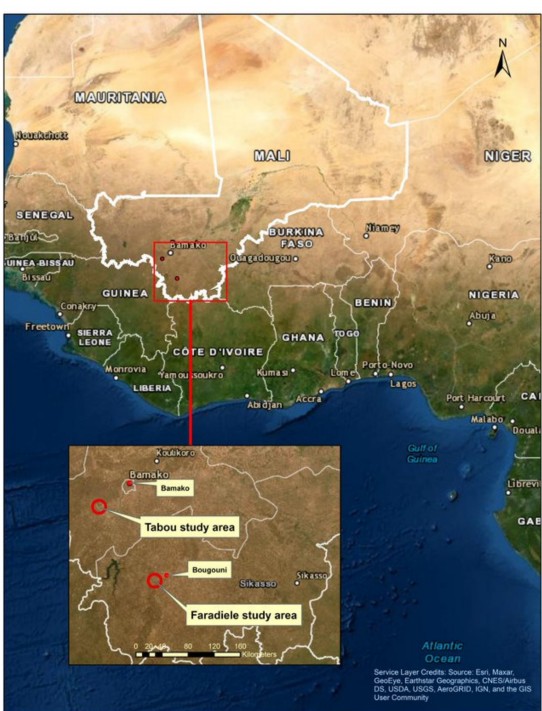

**Figure 5.** Study areas of Tabou and Faradiélé in southern Mali (Figure by S. Winslow).

The vegetation is in the southern Sudanian savanna and is predominantly composed of a mixture of grasses, trees, and shrubs arranged in a complex mosaic. The landscape heterogeneity is a function of underlying soil and hydrology, as well as its agricultural uses, the combinations of which produce unique patterns of land cover (Figure 6) [8,27].

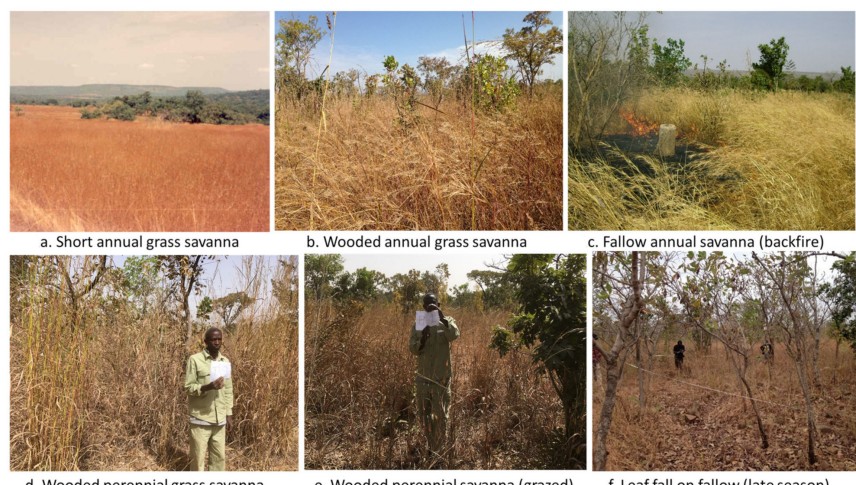

**Figure 6.** Examples of different vegetation types and conditions in Mali (Photographs by P. Laris).

### 2.1. The Local Fire Regime

Fires were set in accordance with the local fire regime (based on a decade of field and remote sensing research) [26,27,34] with the exception of 21 fires set randomly to a variety of plots during the MDS (note that it is not possible to set fires randomly in the EDS, because many grasses are too moist to burn). To recreate the pyrogeography of the study areas, fires were set to specific vegetation types according to our research, which has demonstrated that specific types of grasses typically burn during specific seasons of the year. People set fires in the late afternoon when winds are dying and humidity is rising. They also tend to set backfires, but we designed our experiments to include both head fires and backfires for the purposes of the experiment due to the fact that previous work suggests that fire type strongly influences fire intensity and possibly emission density [4].

### 2.2. Data Collection

Data for the following variables were collected in the field: fire season, average plot biomass, grass percentage of biomass, biomass consumed, fuel moisture, grass species, wind speed, ambient humidity, temperature, fire type, time of day, fire duration, scorch height, and burned area completeness or patchiness. Fuel load (plot biomass) was measured in each of the experimental plots by delineating three representative pre-fire quadrats of $1 \times 1$ m. Grasses were cut at the base using a scythe, weighed with an electronic balance and averaged. When present, leaf litter was weighed separately. When grasses were not fully cured, a sample was cut and weighed wet, then dried and reweighed. The percentage of moisture content was taken as the average for the plot. Vegetation characteristics, including grass type (annual or perennial), grass species, and their height, were recorded for each site based on the sample quadrats.

A Kestrel 5500 Weather Meter station was used to collect wind speed, ambient humidity, and temperature during the burning of each plot. We recorded values every five seconds and averaged them for the entire burn time. The weather station was placed upwind and near each experimental plot 2 m off the ground in an open area. Wind direction relative to the direction of each fire was recorded.

Ignition time was noted and each fire was timed until the flaming front reached the end of the 10 m plot. The majority of fires were set in late afternoon, which is in accordance with local practices, although we set some fires earlier for comparative purposes. Post-fire ash and any unburned material were weighed for areas of similar composition to the $1 \text{ m} \times 1$ m pre-fire quadrats to determine the amount of biomass consumed. Scorch height was averaged for each plot by measuring the height of scorch marks on several small trees. BAC—a measure of the patchiness of the burn—was estimated by two observers.

### 2.3. Plot Design

Plots were selected to represent an array of savanna vegetation types dominated by different grass species and woody cover. To aid in the selection of the burn plots, we used a long-term fire database to select sites with known fire seasonality—that is, fires known to burn during the early, middle, or late fire season on an annual basis [27]. We divided the sites into plots of $10 \times 10$ m and applied treatments of head fires and backfires. Fire timing was set according to the historical pattern of burning with early fires set in November through December, middle fires in January, and late fires in late-February and March (Figure 1). When possible, we conducted multiple burns per site to account for plot-level heterogeneity. Plots at each site were located near each other with attention paid to maintaining consistency in grass type and woody cover. Head fire and backfire plots were located directly adjacent to one another.

### 2.4. Field Data Analysis

To quantify intensity, we used Byram's [35] fireline intensity, which is defined as follows:

$$I = Hwr \tag{3}$$

where I is Byram's fireline intensity (kW/m), H is the net low heat of combustion (kJ/kg), w is the fuel consumed in the active flaming front (kg/m$^2$), and r is the linear rate of fire spread (m/sec$^1$). The net low heat of combustion (H) was selected following Williams [36] with 20,000 kJ/kg as an appropriate value for savanna fires. The load of fuel consumed was calculated by subtracting the average ash and unburned material remaining in three quadrats per plot from the pre-fire measurement of dry biomass. Variable r was derived from the time it took for the first flaming front to reach the end of the 10 m plot. We then calculated fireline intensity, and finally, combustion completeness by dividing the biomass consumed by the pre-fire biomass.

### 2.5. Gas Emissions Sampling and Analysis

We used an IMR 1400 gas analyzer (Environmental Equipment Inc., 3634 Central Ave, Saint Petersburg, FL, USA) to continuously measure gas emissions from the flaming front of each fire. The gas analyzer was calibrated on-site before each use to account for background values of the gases recorded—$CO_2$, CO, and $CH_4$ ($O_2$ and NOx were also recorded but not used in the analysis). During each fire, the nozzle was held approximately 0.5 m above the flaming front and followed the front as it advanced across the plot. Although fast-moving, it was more difficult to collect gas samples from intense head fires than from backfires, and every effort was made to maintain the nozzle position above the flaming front (Figure 7).

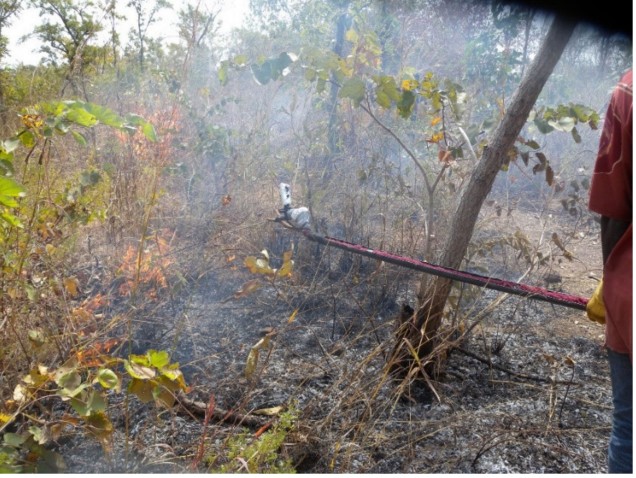

**Figure 7.** Collecting emissions data using a long nozzle on an IMR 1400 gas analyzer from a low-intensity backfire in the village of Tabou (Photograph by P. Laris).

To record gas emissions in real time, we mounted a small video recorder above the screen and recorded values for the entire time of the burn. Fires were also recorded using a small heat-proof camera located on the gas intake nozzle. Backfire plots were lit first in the downwind location from the head fire plot to create a firebreak to prevent the fire from escaping. A 1.5 m firebreak was cut around all experimental plots. Prior to each fire, surrounding areas were wetted using a portable Indian fire pump, which was also used to extinguish any fires burning outside of plot areas. Prior to burning, we conducted an inventory of grass and tree species and classified each savanna plot according to convention. Fires were then lit along a line using a torch to ensure an even flaming front.

Post-fire data processing involved transcribing the recorded values from the video of each fire into a spreadsheet. Post-processing involved removing the first 40% of emissions data to ensure that data were used from the point in which the fire front had fully developed and removing the last 10% to eliminate data on smoldering that typically occurs at the end of each burn. Our objective was to document flaming combustion while recognizing that all savanna fires are a mixture of flaming and smoldering. We then calculated mean values for $CO_2$, CO, and $CH_4$ emissions in ppm for each individual fire to use for the analysis.

Following convention, we calculated EF as follows:

$$EF_x = F_c 1000 \, \frac{MM_x}{MM_{carbon}} \cdot \frac{C_x}{C_T} \tag{4}$$

$EF_x$ is the emissions factor for species $x$ (g/kg). $F_c$ is the mass fraction of carbon in the fuel for which we use the value of 0.5 (the majority of studies find the carbon fraction to vary between 0.425 and 0.50; the latter is most often used for purposes of comparison [37] although Lacaux et al. [38] found a value of 0.425 for West Africa). $MM$ is the molecular mass of species $x$ (g), and 1000 g/kg is a conversion factor. $MM_{carbon}$ is the molecular mass of carbon (12 g), and $C_x/C_T$ is the ratio of the number of moles of species $x$ in the emissions sample divided by the total number of moles of carbon, calculated as follows:

$$\frac{C_X}{C_T} = \frac{ER_{X/CO_2}}{\sum\limits_{j=1}^{n} \left( NC_j ER_{j/CO_2} \right)} \tag{5}$$

where $ER_{x/CO2}$ is the emissions ratio of species $x$ to $CO_2$, $NC_j$ is the number of carbon atoms in compound j, and the sum is over all carbonaceous species (approximated as $CO_2$, CO, and $CH_4$ for this study).

### 2.6. Statistical Analysis

Prior to the statistical analysis, outliers in the dataset were removed using the quartile method. This left 86 records in the dataset for correlation analysis. After examining the histograms for each variable, we conducted *Log* or square transformations to bring irregularly distributed variables closer to normal distribution and to improve scedasticity. We ran ANOVA tests to look for significant differences in the results for $CH_4$ EF and ED by season. We used bivariate regression analysis to look for correlations between the two dependent variables—methane EF and density—and independent variables, which included Byram's fire intensity, percentage of grass biomass, fire spread rate, total fuel moisture, combustion completeness, burn patchiness, wind speed, ambient temperature, and humidity. We normalized values for Byram's fire intensity, spread rate, and scorch height by head fires and backfires because these three factors are strongly influenced by fire type and there was an unequal number of head fires and backfires in some instances. Below we present data in the original (with all outliers) as well as modified forms when necessary. All of the statistical analysis was conducted using JASP software, version 0.1.3.

## 3. Results

### 3.1. General Fire Characteristics by Season

The results for the mean plot and general fire characteristics by study period and fire type are shown in Table 1. The importance of including the MDS (as opposed to the binary EDS/LDS) is apparent. In terms of ambient weather conditions, the average temperature generally increases over the dry season, but there is a dip mid-season. Conversely, the mean wind speed peaks mid-season, although the wind speeds are relatively low (mean = 1.2 m/s). The average humidity decreases as the dry season progresses (although the MDS had the greatest variation in values due to occasional mango rains). The fuel moisture peaks in the MDS due to the burning of uncured perennial grasses. In terms of biomass, the percentage of grass in the total plot biomass is greatest in the EDS (91), while the total dry biomass is higher in the MDS (4.19 t/h), reflecting an increase in leaf litter (lower grass percentage) as the dry season progresses. The total biomass declines slightly in the LDS probably due to animal consumption of grassy biomass.

**Table 1.** Mean Plot and General Fire Characteristics by Study Period and Fire Type for all Fires in Tabou and Faradiele, Mali, 2015–2016.

| Mean Plot Characteristics (*n* = 107) | Annual Mean | Head (51) | Back (56) | Early (25) | Middle (42) | Late (40) |
|---|---|---|---|---|---|---|
| Dry biomass (tons/hectare) | 4.1 | 4.0 | 4.2 | 3.8 | 4.2 | 4.1 |
| Grass biomass (percent) | 78.1 | 76.1 | 80.0 | 91.1 | 72.5 | 75.9 |
| Temperature (Celsius) | 33.3 | 33.3 | 33.3 | 32.8 | 31.2 | 35.8 |
| Relative humidity (percent) | 22.1 | 22.1 | 22.7 | 29.1 | 23.9 | 15.9 |
| Wind speed (meters/second) | 1.19 | 1.18 | 1.20 | 1.01 | 1.45 | 0.86 |
| Spread rate (meters/second) | 0.030 | 0.043 | 0.018 | 0.032 | 0.026 | 0.034 |
| Byram's Fire Intensity (Kw/m) | 214.1 | 314.9 | 124.5 | 229.6 | 179.5 | 249.7 |
| Scorch Height (meters) | 1.5 | 1.7 | 1.3 | 1.4 | 1.3 | 1.7 |
| Fuel Moisture (%) | 8.9 | 10.1 | 7.9 | 8.6 | 14.3 | 3.5 |
| Burn Area Completeness (%) | 91.8 | 93.1 | 90.7 | 84.2 | 90.4 | 98.0 |
| Combustion Completeness (%) | 85.6 | 85.8 | 85.4 | 83.7 | 83.2 | 89.5 |

The characteristics of the fires also vary by season. The value of Byram's fire intensity was the highest in the late season (249.7 kW/m), as expected, but surprisingly slightly lower mid-season compared to the early season (179.5 Kw/m as compared to 229.6). Note that moist grasses (perennials) do not burn in the EDS, only dry, fine annual grasses do. As a result, the EDS tends to have lower fuel moisture and a higher intensity than the MDS. This might seem counterintuitive, but recall the study design is based on replicating local burning practices, not systematically studying the impact of seasonality; as such, perennials with higher moisture contents were burned in the MDS. In addition, we found a large variation in the fireline intensity, especially for head fires and fires set in the middle of the day. The calculated intensity values ranged from 12 to 1395 kW/m for all the plots. While the minimum intensity increased over the fire season, the maximum intensity decreased. Similarly, scorch height, a good proxy for the fire intensity, followed the same pattern with the LDS fires being the highest and the MDS fires being the lowest, with the EDS value in between. The fire spread rate, a key determinant of intensity, had a similar pattern with a speed higher in the LDS compared with the EDS, and with the MDS being the lowest despite the stronger winds in the latter. As can be seen, the fire type has a large influence on fire intensity; the head fire mean intensity was much greater than that of the backfires, as expected (314.9 kW/m compared to 124.5 for head fires).

The BAC and CC generally increased from EDS to MDS to LDS as expected due to fuel drying. The mean BAC increased as the dry season progressed to a near complete burn by the late season (84.2% to 90.4% to 98.0%) while the percentage of biomass consumed (CC) dipped slightly from the early to middle season (likely due to fuel moisture effects) but increased significantly in the late season (83.7% to 83.2% to 89.5%). The total combustion efficiency (BAC * CC) steadily increased from 71% in the EDS to 77% in the MDS to 82% in

the LDS, indicating that a greater percentage of fuels burned as the dry season progressed as expected.

*3.2. Emissions by Season*

In terms of the parameters of the gaseous emissions, we found the mean $CH_4$ EF was 6.96 g/kg and that the $CH_4$ EF was the lowest in the LDS and the highest in the MDS, with the EDS falling in between (Table 2). We found that the methane density did not increase over the course of the dry season—it rose from 2.15 ($g/m^2$) in the EDS to a high of 2.87 ($g/m^2$) in the MDS before dropping to 1.72 ($g/m^2$) in the LDS with an average of 2.27 ($g/m^2$). The overall ANOVAs for $CH_4$ EF and ED are quite significant ($p = 0.002$ and 0.074, respectively), indicating significant contrast among at least one of the three comparisons. The EDS and LDS are not significantly different from one another but both are significantly different from the MDS (see Supplementary Materials). We also found the backfires had a lower methane density than the head fires (2.16 $g/m^2$ and 2.39 $g/m^2$, respectively). However, when we removed the outliers, the pattern reversed, and the values were substantially lower, thus a few intense head fires skewed the results. This is not surprising, given that head fires with higher winds can have much higher intensity values, which can affect emissions. As seen in Table 2, the greatest effect of removing the outliers was on the values for the headfires; the backfire values did not change much. Overall, the backfires had lower $CH_4$ EFs than the head fires (6.80 g/kg to 7.12 g/kg); this difference was the strongest in the MDS, and the pattern was reversed for the EDS, which had lower head fire emissions. Somewhat surprisingly, the MCE values declined over the course of the dry season, probably driven by the increase in leaf litter and the decline in winds in the LDS (it was expected the MCE would rise due to the lower fuel moisture in the LDS). The backfire MCE was higher than that for the head fires, but this too was influenced by the outliers. When the outliers were removed, the MCE for the head fires increased from 88.6 to 92.4 (shifting from smoldering to flaming), while the removal of outliers did not change the backfire MCE (see below). Backfires cannot burn moist grasses (thus the reason for higher fuel moisture values for head fires as opposed to backfires) and this may explain the differences in the results for the MCE. Finally, when we removed the "randomly" set fires from the MDS data, the methane EF and ED both increased. This increase was driven by much higher methane EF values for the locally set head fires, which were nearly double those of the local backfires (13.7 g/kg to 6.99 g/kg), likely the outcome of higher fuel moisture levels for the locally set fires.

**Table 2.** Methane Emission Parameters and Key Fire Metrics by Fire Season and Type for All Fires in Tabou and Faradiele, Mali, 2015–2016 (EF is emission factor, MCE is modified combustion efficiency, BAC is burned area completeness, CC is combustion completeness, and kW/m is kilowatts per meter).

| Emissions and Fire Data $n = 107$ Values in Parentheses Have Outliers Removed ($n = 86$) | CH4 Density g/m² | CH4 (EF) g/kg | MCE | Total Combustion (BAC + CC) | Byram's Fire Intensity kW/m |
|---|---|---|---|---|---|
| **All Fires (mean)** | 2.27 (2.09) | 6.96 (6.93) | 90.1 (91.1) | 0.77 (0.78) | 224.5 (165.6) |
| **Early Fires** | 2.15 (1.60) | 6.52 (5.90) | 94.9 (93.1) | 0.71 (0.69) | 229.6 (154.5) |
| **Middle Fires w/random burns** | 2.87 (2.73) | 9.53 (9.76) | 89.5 (95.1) | 0.77 (0.75) | 162.2 (138.8) |
| **Middle Fires wo/random burns** | 3.16 (3.14) | 8.88 (10.35) | 85.0 (93.8) | 0.77 (0.75) | 170.9 (117.9) |
| **Late Fires** | 1.72 (1.82) | 4.53 (4.93) | 87.7 (87.7) | 0.82 (0.87) | 249.7 (199.5) |
| **Head Fires** | 2.39 (1.86) | 7.12 (6.86) | 88.6 (92.4) | 0.79 (0.79) | 296.4 (232.2) |
| **Backfires** | 2.16 (2.24) | 6.80 (6.98) | 91.5 (91.6) | 0.75 (0.78) | 120.1 (119.8) |

### 3.3. Regression Analyses

Our objective was to explain the factors that cause higher $CH_4$ EF and ED values. As such, we regressed numerous variables against these two dependent ones. Using the raw data, we found that only wind speed and humidity had significance for both the forward and backward regression models for $CH_4$ EF. Both models were highly significant but with weak effect sizes ($R^2_{adj}$ of 0.215). In terms of the $CH_4$ ED, the backward model found significance for the Byram's fire intensity, fire rate of spread, and wind speed, resulting in a highly significant model, but one with a weak effect size ($R^2_{adj}$ of 0.205). The forward model kept only the wind speed, was also highly significant, but was trivial in its effect size ($R^2_{adj}$ of 0.122). We performed a second regression after transforming the variables with a non-normal distribution to produce the results for the *log* of $CH_4$ EF and ED values. The results were similar for the *log* of $CH_4$ EF, but the wind speed was the only significant variable (the humidity dropped out). While significant (F = 13.249, $p < 0.001$), the effect was feeble ($R^2_{adj}$ of only 0.153). In terms of the *log* $CH_4$ ED, the model found the wind speed, *log* Byram's intensity, and *log* fire spread rate to be highly significant (F = 9.569, $p < 0.001$). That said, again the effect size was weak ($R^2_{adj}$ = 0.274), thus only about a quarter of the variability in the *log* $CH_4$ ED was accounted for by the combination of these three variables.

In general, we found only a few of the variables were significantly correlated with the $CH_4$ EF or ED, although the trends tended to be consistent across the fire types and seasons with some key exceptions by season. We found the $CH_4$ EF was positively correlated with the MCE with a high significance for "all fires", backfires, and MDS fires ($p < 0.01$) and with a low significance for the head fires, the EDS, and the LDS (Figures 8 and 9, Tables S1–S6). We found the $CH_4$ EF was positively and significantly correlated with wind speed for all the fires ($p < 0.001$) and backfires ($p < 0.01$), but not for the head fires. The only other variable significantly correlated was the ambient temperature, which correlated negatively with the $CH_4$ EF for all the fires and head fires ($p < 0.05$) and with a low significance for the backfires. There were no other significant relationships when the $CH_4$ EF was analyzed by fire season. We did find that the EFs for CO and $CH_4$ were negatively correlated. That is, the $CH_4$ EF did not increase with the increasing EF CO (decreasing MCE) as is commonly thought; rather, the $CH_4$ EF was positively and significantly correlated with the MCE for all the fires, backfires, and MDS fires and trended positive for all the fire types and seasons.

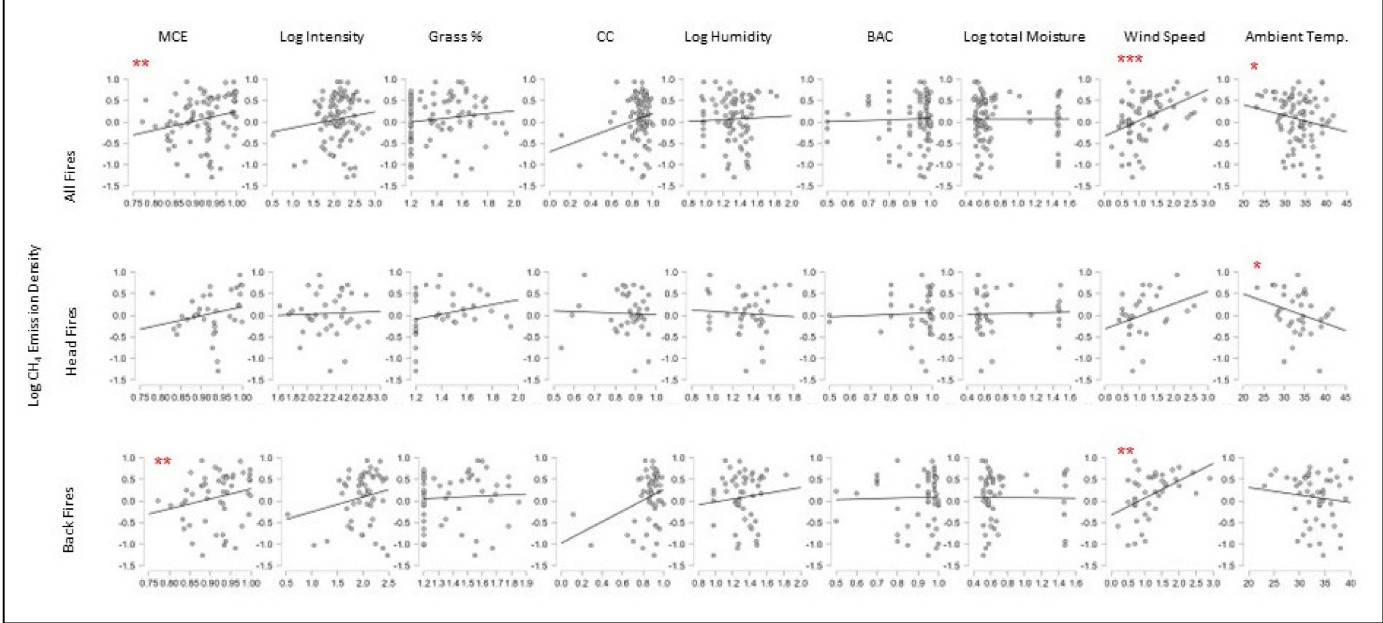

**Figure 8.** Methane Emission Factor (*Log*) Correlation Plots for Key Variables by Fire Type (\*\*\* $p < 0.001$, \*\* $p < 0.01$, \* $p < 0.05$).

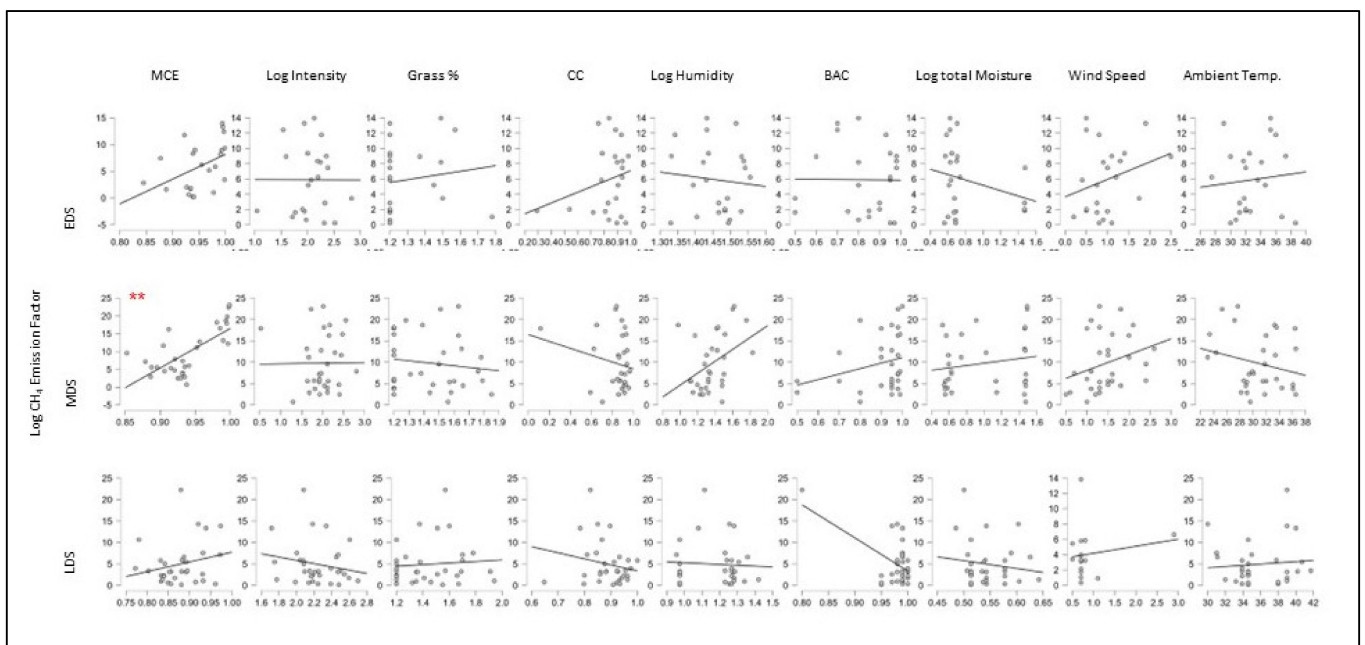

**Figure 9.** Methane Emission Factor (*Log*) correlations for key variables by fire season (\*\*\* *p* < 0.001, \*\* *p* < 0.01, \* *p* < 0.05).

As noted, there were trends in the data that were consistent across the categories of fire type and season, although the results were not statistically significant. For example, we did not find major differences in the trends for head fires and backfires in terms of the variables that correlated with the EF CH$_4$ (Figure 8), which was somewhat surprising given the differences in fire intensity, flame height, and speed for these two types of fires (Table 1). In terms of weather conditions, we found the wind speed and humidity trended positively with the EF CH$_4$ for fire type and season while the ambient temperature trended negatively.

Other variables trended in different directions when the results were compared by season. For example, the fire intensity had a positive correlation with the EF CH$_4$ for the MDS but was negative for the other seasons. The CC correlated positively with the EF CH$_4$ in the EDS, but slightly negatively in the other seasons. The grass biomass percentage showed slightly different trends: in terms of fire type, with the head fires being negatively correlated and the backfires positively correlated; and in terms of season, with the EDS being positively correlated and the LDS being negatively correlated with the EF CH$_4$. None of these findings had significance (Figures 8 and 9). We discuss potential reasons for these shifts by season below.

In terms of emission density relationships (Figures 10 and 11; Tables S6–S12), the MCE (*p* < 0.05), CC (*p* < 0.05), and wind speed (*p* < 0.001) all correlated positively with the methane ED for all the fires. The grass biomass percentage was negatively correlated with the methane ED, but insignificantly. The backfire ED showed a positive and significant correlation with the wind speed (*p* < 0.01) and CC (*p* < 0.05). The head fires had no significant correlations, although similar positive trends were observed for wind speed and the MCE, and negative trends were observed for the grass biomass for all fires. Head fires also had a negative (although insignificant) relationship between the CC and methane ED, the opposite of that for backfires. The humidity and fuel moisture did not influence the ED regardless of fire type while ambient temperature was negatively, but insignificantly, correlated with ED for all the types.

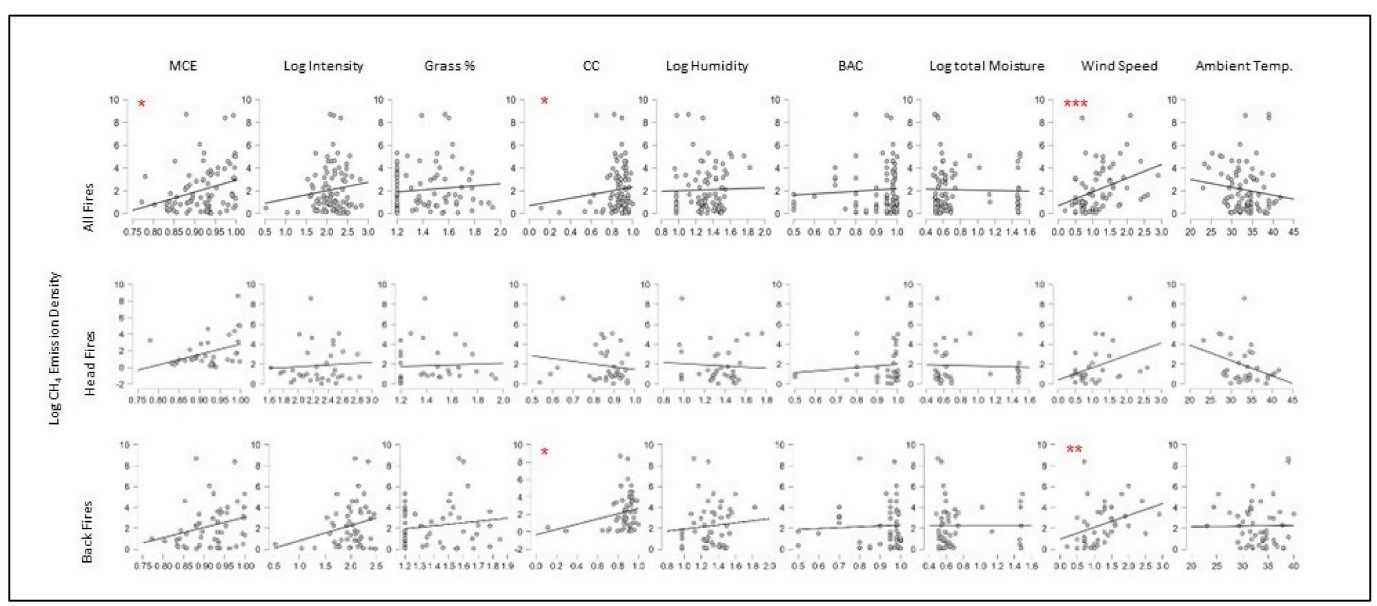

**Figure 10.** Methane ED (*log*) Correlation Plots for Key Variables by Fire Type (*** $p < 0.001$, ** $p < 0.01$, * $p < 0.05$).

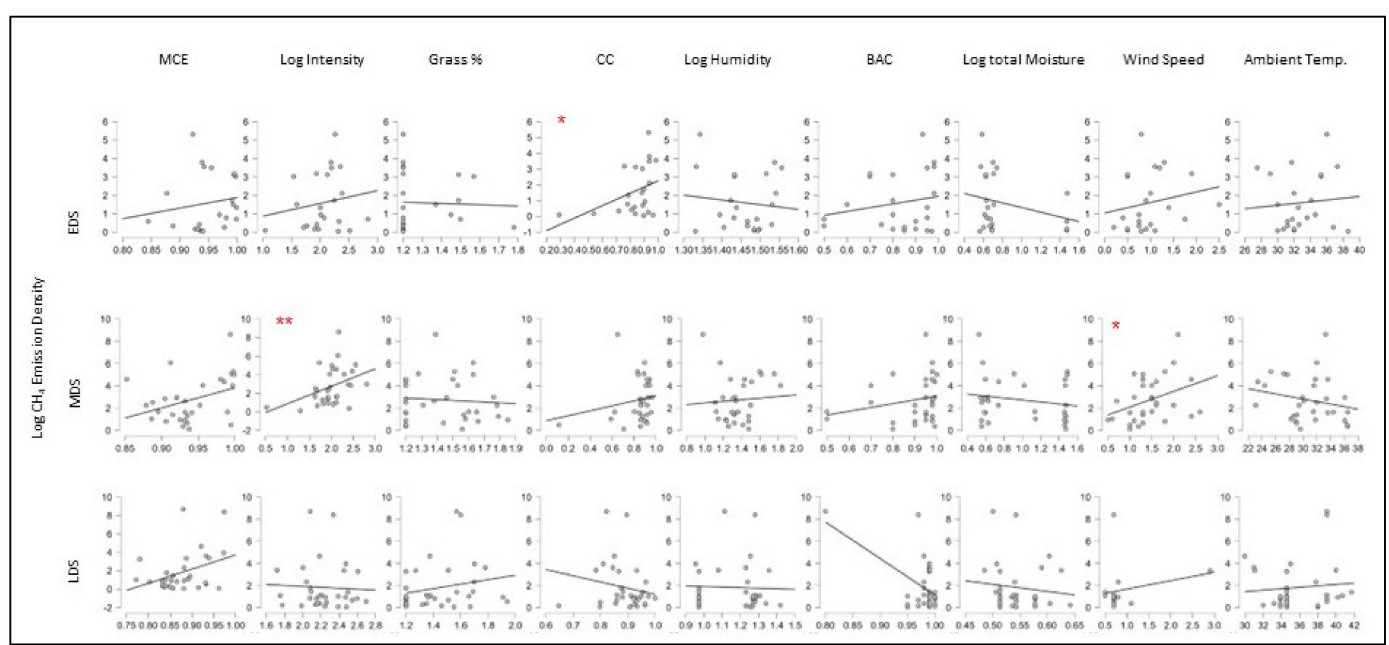

**Figure 11.** Methane ED (*log*) Correlation Plots for Key Variables by Fire Season (*** $p < 0.001$, ** $p < 0.01$, * $p < 0.05$).

When disaggregated by season, we find that the EDS and MDS show similar trends for the $CH_4$ ED that are distinct from the LDS. The EDS and MDS had similar trends for the $CH_4$ ED, which increased with the MCE, intensity, CC, and wind speed, while the LDS results indicate that the methane ED decreased with intensity and the CC (not significantly). Surprisingly, the methane ED in the late season does not correlate positively with the CC.

We also found that for "all fires", the EF CO and EF $CH_4$ had a negative and highly significant relationship (Figure 12 and Table S13). We compared trends for ambient weather conditions and found that, in all cases, the EF CO and EF $CH_4$ trended in opposite directions. That is, increasing winds and humidity tended to increase the EF $CH_4$ and decrease the EF CO, while the opposite was true for increasing temperatures. We also found that as the percentage of grass rose, the EF CO fell while the EF $CH_4$ rose (although not significantly).

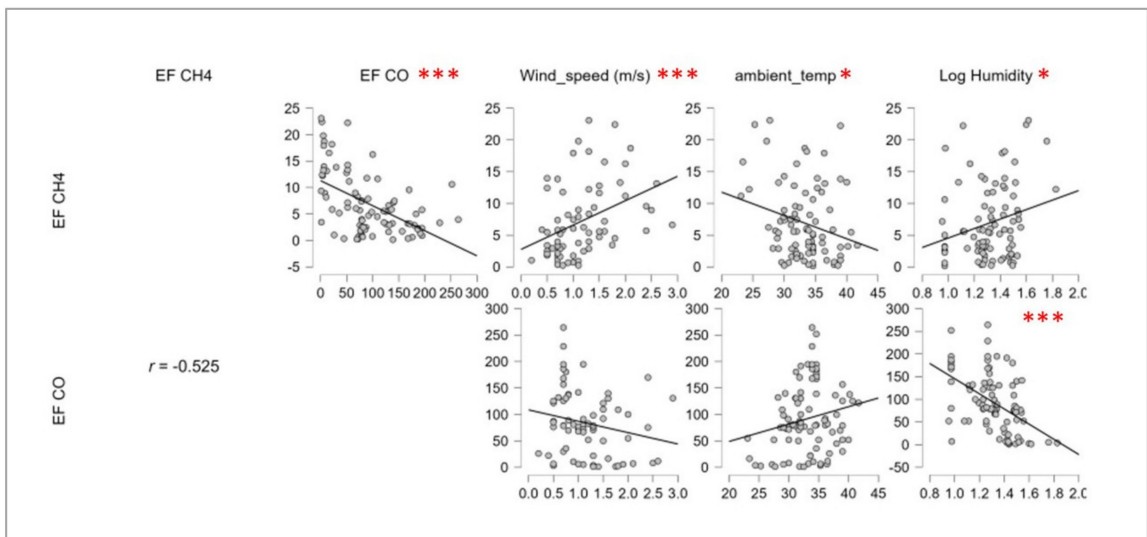

**Figure 12.** Correlation Plots for Methane and Carbon Dioxide EF and Ambient Weather (*** $p < 0.001$, ** $p < 0.01$, * $p < 0.05$).

*3.4. Comparison of Local and Random Fires in the MDS*

When comparing our results for randomly set fires in the MDS with those set according to local practices, we found the "local" fires had a higher $CH_4$ EF than randomly set ones. The higher $CH_4$ EF values were a function of much higher head fire EFs for local fires. The local backfire $CH_4$ EF was 7.2 g/kg compared with 10.6 g/kg for the head fires. The difference between the head fires and backfires for the random fires was much smaller (Table 2). In terms of the regressions, the local fire $CH_4$ ED had a positive and significant correlation with intensity ($p < 0.01$), while the random fires were positively and significantly correlated with the CC ($p < 0.05$). The grass biomass percentage was positively correlated with the ED for the local fires and negatively for the random fires, although the results were not significant. Importantly, the total moisture (*Log*) was significantly and negatively correlated with the local ($p < 0.05$) fires, but not with the random fires. Otherwise, the trends were similar and mostly with a low significance (Figure 13, Tables S14 and S15).

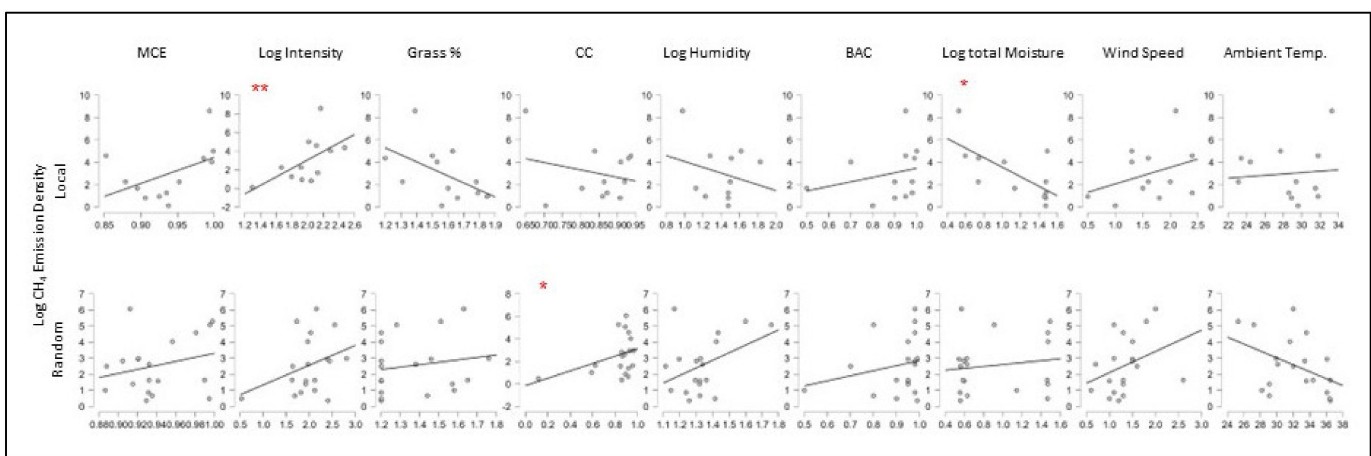

**Figure 13.** Methane ED Correlation Plots for Key Variables for Local and Random Fires (*** $p < 0.001$, ** $p < 0.01$, * $p < 0.05$).

## 4. Discussion

*4.1. Causes of Elevated Methane Emission Parameters and Seasonal Differences*

The study found significant variations in the parameters of methane emissions as well as in the relationships between the key factors thought to affect the $CH_4$ EF and ED

for different seasons. Many of these findings did not support the theoretical postulates presented above. We had expected the CH$_4$ EF to peak in the EDS and the CH$_4$ ED to peak in the LDS, but this was not the case. Critically, the changes in the CH$_4$ EF and ED were not linear over time as expected—both were at a maximum in the MDS and a minimum in the LDS. We argue that this was due in part to the local practices of seasonal mosaic burning, which complicates the analysis of the relationship between the key variables and methane emissions. As noted, this is largely because people set fires to specific vegetation types at specific points in the dry season and the conditions of the fuels are therefore a function of human practices as much as natural seasonality.

This study finds that fires set in the MDS differ from those set in the early or late dry season, supporting our argument against binary EDS/LDS emission models and policies. We found the EDS and LDS were not significantly different from one another, but both were significantly different from the MDS. In addition, MDS fires have a lower intensity than both EDS and LDS fires, while the CC varied little from the early to middle season but increased substantially by the late season. The BAC increased steadily over the course of the dry season and the mean values for the total burn efficiency gradually increased from the early to the middle to the late season as expected due to the gradual drying of the biomass. Somewhat surprisingly, the MCE declined during the dry season due to a variety of factors—we suspect the higher fuel moisture caused a decline in the MDS and the higher leaf litter in the LDS. The fuel loads increased in the middle and late dry seasons as did the percentage of leaf litter in the total biomass, as expected. These factors, as well as the changes in the weather conditions by season, had differing effects on the methane emission parameters. We expect that these differences explain the low explanatory power of the regression models.

We found that the increasing fire intensity resulted in higher CH$_4$ EF values for the MDS, while the opposite was true in the EDS and LDS. This finding is most likely because the leaves were burned while green on trees in the MDS (higher intensity fires have taller flames), while the fires in the EDS burned areas with little leaf matter and those in the LDS burned dry, fallen leaves (see below). The findings from our study also indicate that the CH$_4$ EF rose with the MCE (EF CH$_4$ was not coupled with EF CO as is often suggested in the literature [39]. Indeed, the regression analysis indicated that not only are the EF CO and CH$_4$ negatively and significantly correlated for our study area, but the key factors affecting combustion efficiency, such as weather conditions (especially wind speed), have differing impacts on them. The EF CH$_4$ decreases with the increasing CC while the EF CO increases; this finding was expected since the two key factors causing the lower CC are typically increasing humidity and fuel moisture, both of which result in higher EF CH$_4$ and lower CO values. Increasing the wind speed, which also tends to increase the CC (especially for backfires), lowers the EF CO (Figure 12). Finally, we also found that as the percentage of the grass in the fuel rose, the EF CO decreased while the EF CH$_4$ increased (although not significantly). In other words, as the woody fuels shifted from green leaves on trees to dead leaf litter, the CH$_4$ emissions fell, even as the CO emissions rose (although we cannot tell from the data the amount of tree cover, only the leaf litter as a percentage of the total fuels).

In terms of the regression analysis, we found that the models for both the *log* CH$_4$ EF and ED had a weak effect size, with only the wind speed having significance for the *log* CH$_4$ EF and only the Byram's intensity, wind speed, and fire speed having significance for the *log* CH$_4$ ED. That said, it is worth noting that Byram's intensity is a function of fire speed and that wind speed is a key determinant of fire speed (especially for head fires), thus wind speed is an important determinant of both CH$_4$ EF and ED. It is also critical to note that while many savanna fire studies are based on head fires conducted under windy conditions, most intentional fires set by local users are as backfires set in the afternoon when winds are falling [14].

In summary, these data suggest a very complex relationship between fuel composition, fuel state (green leaves on trees vs. dead leaf litter), fuel moisture content, and ambient weather. The fire type appears to have less of a direct impact on the CH$_4$ EF than we

found in our previous work [4]; however, we did find that the fire type was important in specific circumstances. The backfires had higher $CH_4$ EF values in the EDS, and the head fires had higher $CH_4$ EF values in the MDS, with no difference in the LDS. We found that increasing the fire intensity resulted in higher CC and BAC values and thus higher $CH_4$ ED values, especially for the backfires. We suspect this is due to the increased airflow, which is critical for combustion completeness and fire spread in backfires. We also found that the fire intensity correlates with the increasing $CH_4$ ED for all seasons and fire types. This is primarily because the increasing intensity causes a rise in the total burned percentage (CC * BAC) (especially for backfires) and not necessarily because of the effect on the $CH_4$ EF, which did not correlate with rising intensity except in the MDS.

Importantly, we found the $CH_4$ ED was positively correlated with several of the key fire variables, including fire intensity and CC (although not significantly in all cases). This relationship holds for both backfires and head fires but is stronger for backfires. In general, the effects of key variables, such as wind, humidity, FM, and intensity, on the CC and BAC are stronger for the backfires than head fires, which is not surprising given that head fires tend to have a high degree of variance and are difficult to model [40].

We found increasing the wind speed was associated with a higher fire spread rate and a higher fire intensity, which elsewhere we found to increase the $CH_4$ EF [4]. In this study, however, we found that increasing the fire intensity correlated negatively, but insignificantly, with the $CH_4$ EF for all the fire types ($CH_4$ EF increased with fire intensity when the outliers were included, although not significantly and the removal of outliers reversed the trend). However, when disaggregated by season, we found that during the MDS, the fire intensity correlated positively with the $CH_4$ EF but that this relationship was negative during the EDS and LDS, although not significantly. We also found that the MCE was positively correlated with the $CH_4$ EF during all seasons, with only the MDS having a significant relationship. The wind speed was also positively and significantly correlated with the $CH_4$ EF during the MDS. These findings suggest that *in the MDS, elevated wind speeds affect burning in such a way as to increase both the methane emissions and MCE*. In general, we found wind speed and humidity to trend positively with $CH_4$ EF (note that both were highest in the MDS) for fire type and season while an ambient temperature trended negatively. Finally, in terms of the fire type, the head fires had a negative (although insignificant) relationship between the CC and $CH_4$ ED, the opposite of that for the backfires.

Another key finding was that the $CH_4$ EF did not vary positively with the CO EF; indeed, as the CO EF fell, the $CH_4$ EF increased significantly. We suggest that the low combustion efficiency (at the molecular scale) results in a higher CO EF value, as expected, but a lower $CH_4$ EF value. Contrastingly, a lower CC (at the patch scale) results in higher $CH_4$ and lower CO EF values. As such, it appears that methane emissions are higher with the increasing MCE (molecular) and lower with the increasing CC (patch level). These results suggest a complex relationship between fuel combustion efficiency and completeness, which can be related to fuel composition, state (green or brown leaves), fuel moisture content, and other potential factors. Fire type appears less important than in our previous work for determining EF $CH_4$ [4]. However, increasing fire intensity does cause higher methane ED for the reasons explained above.

Our field observations suggested that the raw methane emissions measured in ppm spiked when small trees with green leaves combusted along the fireline. Because our methodology was to "follow the flaming fire front" and, because small trees and shrubs burn more slowly than grasses, we postulate that our study over-sampled the burning of live shrubs and tree combustion when compared with grass and leaf litter. This, we believe, resulted in relatively high mean $CH_4$ EF values, and might also explain, in part, why the CO and $CH_4$ were not positively correlated.

The raw data are shown in Figure 14 in ppm for both the CO and $CH_4$ over time during a single burn of an experimental plot (Plot 1-2-8). Note that initially the emissions of $CH_4$ were steady but were a small fraction of those for CO, which was expected. Then, at time t = 4130, the emissions of $CH_4$ spike preceding a similar spike in CO. Figure 14b

shows plots of CO vs. CH$_4$ for the same fire and illustrates the "looping" pattern of the emission ratio. The observed changes in the CH$_4$/CO ratio are associated with different phases of burning, such as ignition, flaming, glowing, and smoldering (Yokelson, personal com). The data from Yokelson's lab experiments demonstrate that the ratio of CH$_4$ to CO rises linearly only during the initial phases of burning; it then declines during the later stages of combustion. By maintaining our gas collection nozzle over small trees throughout the combustion process, we likely oversampled the latter phases of the combustion of small trees and shrubs when the methane emissions are elevated.

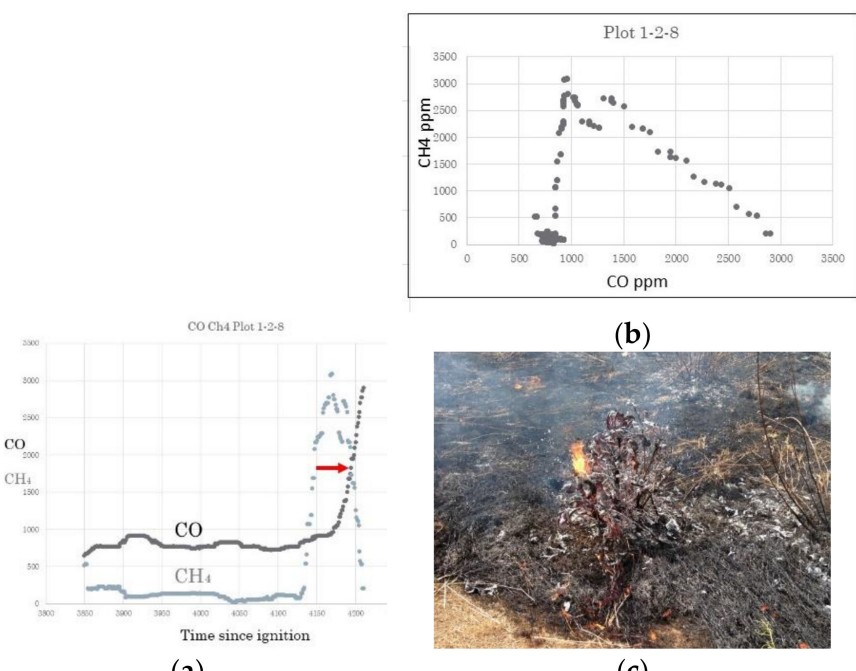

**Figure 14.** Relationship between CO and CH$_4$ emissions over time during a savanna fire in Mali with small trees. (**a**) Changes in CO and CH$_4$ emissions over time. (**b**) Changes in the CH$_4$ to CO ratio during the burning of a small tree. (**c**) Final stage of burning of a small tree; note that the flaming front has advanced well ahead in the grasses.

We conclude that our values for CH$_4$ EF should be interpreted with caution, as it is likely that we overestimated the methane emissions from the burning of green leaves on shrubs and small trees for our individual plots; however, it is also possible that we captured a critical phenomenon that produces elevated methane emissions. It should also be noted that Wooster [41] found that the ground values of CH$_4$ EF were higher due to "sampling a greater proportion of smoke from smoldering processes than is generally the case with methods such as with airborne sampling" (11592). Wooster et al.'s work was one of a few other studies measuring gases at ground level and perhaps they also captured some of the processes we illustrate in Figure 14. They also report residual smoldering CH$_4$ EF values above 7.0 for several fires, which is on par with our results. As Akagi and colleagues [42] (2011) note, nearly all measurements of savanna fire emissions have been performed using low-level airplane flights. Furthermore, as Yokelson and colleagues [39] argue, comparisons of nascent and downwind smoke samples reveal intriguing post-emission changes in smoke composition due to photochemistry and cloud processing, demonstrating the need to measure smoke less than a few moments in age to properly determine the "initial emissions" from fires. To our knowledge, few savanna studies have done this. Moreover, most savanna emissions studies are from grass-dominated landscapes with far fewer trees than are found in many mesic savannas which have significant canopy cover and large numbers of juvenile trees. Indeed, this was the finding of Nisbet and colleagues [43] (2021), who used isotopic data to conclude that the methane emissions from

fires in West Africa were *primarily* from woody fuels and leaf litter. As such, one might expect, therefore, that emissions from wooded savannas, such as the ones that are common in West Africa, might have emission factors more in line with dry forests, which is indeed what we found.

### 4.2. Local (Systematic) vs. Random Fire

The differences between the locally (systematically) and the randomly set fires described above can be explained in part by the fact that the local fires burned fuels with higher FM than the random fires (16.8% compared to 12.2%). This result was expected because people aim to burn fuels before they are completely dry [26,27]. The local fires also had lower MCE (0.90 to 0.94) values and the local head fires had a dramatically lower MCE of 0.81, which falls well within the smoldering range (again, not surprising given the high fuel moisture in the MDS). It is noteworthy that some grasses simply could not carry a fire when lit as a backfire in the MDS; the fuel moisture was simply too high. The grass biomass percentage was also much lower for the local fires (62%) which means there was more leaf litter than for the random fires (81% grass). This finding suggests that local fires are set in areas with higher tree coverage in the MDS because more leaf litter is associated with a greater number of small trees. Note that although leaf litter peaks in the LDS, leaf fall begins in the MDS. In addition, as noted, the total moisture and $CH_4$ ED were significantly and negatively correlated with the local but not with the random fires, suggesting that the factors governing fires set in the traditional manner by people differ from those set randomly. Otherwise, the trends were similar and with mostly a low significance.

### 4.3. Policy Implications

This study found that the MDS fires had the highest methane emission parameters in terms of both $CH_4$ EF and ED, while the late fires had the lowest values. These findings cast doubt on the proposed approach by [20] which posited methane emissions can be reduced by simply shifting fires to earlier in the dry season. Indeed, the results indicate that shifting burning from the LDS to MDS would result in a doubling of the $CH_4$ EF, which would override any changes in total amount of fuels combusted; the MDS methane ED is 2.87 ($g/m^2$) compared to 1.72 ($g/m^2$) for the LDS (2.73 to 1.82 with the outliers removed). While theoretically shifting the areas burned from the MDS to the EDS would reduce the ED (2.87 to 2.15 $g/m^2$), in reality this is not possible given that many grassy fuels are not dry enough to burn in the EDS. Indeed, it would be impossible to burn them with a backfire as is customary and safer. Head fires burning moist grasses and leaves would undoubtedly result in further increases in $CH_4$ EF and more importantly would not be acceptable to local fire users because head fires are more difficult to control.

The study also found that the local practice of burning grasses in the MDS (when they are just dry enough to burn) resulted in higher $CH_4$ EF and ED values (3.00 $g/m^2$ to 2.56 $g/m^2$) when compared to randomly set fires. However, when only backfires are considered for local burning (the most common practice) do local fires produce less methane ED than random fires (2.27 $g/m_2$ to 2.56 $g/m^2$). This finding indicates that the traditional backfire burning of green perennial grasses in the MDS emits higher levels of methane than those set later in the LDS, but that locally set backfires produce less methane than head fires would for the same vegetation type. We suspect this is because backfires burn far fewer green leaves on trees (below). It should also be noted that a recent study of fire directionality found that most fires in the West African savanna region (excluding the arid Sahel) burn in an E–NE direction, which is contra-wind and is in agreement with our previous interview findings [44].

It is important to put our findings on methane emissions within the context of the seasonal mosaic burning regime we previously documented for the study areas (Laris 2011). According to our research, the peak in fire setting is in the MDS, while the largest amount of area burned occurs in the EDS and the lowest amount of area burned is in the LDS with the MDS falling in between. For example, we found the mean values of the

percentage of coverage burned to be 32.5%, 20.4%, and 5.8%, respectively, for the EDS, MDS, and LDS fires. Our research also found that the burn patterns in the EDS have the greatest fragmentation levels followed by the LDS and lastly the MDS. In addition, our previous work demonstrated a correlation between fire regime and land cover. Specifically, places with a greater area of short annual grasses (short-grass savanna) have earlier burning times than those with less-short-grass savanna. Similarly, areas with highly fragmented patterns of short-grass savanna tend to have a greater burn heterogeneity throughout the fire season. In summary, the vegetation pattern—especially grass species' distribution—is a key determinant of the spatiotemporal pattern of fires [12,45]. This has important implications for policies. It would not be feasible to shift the burning regime earlier in areas with little short-grass savanna even if desired, because the taller perennial grasses cannot be burned in the EDS. Moreover, as our data also show, even if it was possible to shift the fire regime earlier, the burning of moister perennial grasses and tree leaves would likely produce high levels of methane emissions especially if head fires were necessary to carry the flame. We thus conclude that the "where" and "what" of burning (pattern and type of vegetation) and not simply the "when" (time of year) determines the fire regime and ultimately the methane emissions. That is, pyrogeography and not simply fire timing determines methane emissions in a savanna landscape.

In summary, while we agree in principle with [20] policy that widespread EDS burning is critical for savanna and emissions management, we argue that this model works precisely because people select appropriate patches to burn as early as possible. The burning of fine annual grasses with low fuel moisture and tree cover results in the low $CH_4$ EF and ED values we recorded for the EDS and fragments the landscape to limit the spread of later fires. In sum, the current widely used practice of seasonal mosaic burning [26,27] may very well result in lower methane emissions than a more random fire regime, but the reasons for this have less to do with the *timing* of fire than with *geography*. We find it ironic that policymakers are once again attempting to coerce people to alter their fire regimes in the name of reducing environmental damage (e.g., [17]) without adequate research on and knowledge of local burning practices, which are necessary for developing good policies. The recent events in Darwin, Australia, where the policy originated, are telling; researchers found an increase in hazardous smoke pollution, an outcome of less complete combustion, after an EDS policy was put into place to reduce methane emissions [46].

**5. Conclusions**

This study finds that multiple factors interact in complex ways to determine the methane emissions from savanna fires. We conclude that the pyrogeography—the where and when of what is burned—is critical in determining the quantity and types of gases emitted from fires. In many instances, the trends in methane emissions varied by season, although not necessarily as expected. This was more pronounced for the emissions density than emissions factors. We found that the $CH_4$ EF and ED both reached their maximum during the MDS. The peaks in the methane emission parameters were likely the result of higher fuel moisture levels of the perennial grasses burned in combination with the higher winds and the greater amounts of green tree and shrub leaves burned during this season. It is important to note that the highest methane emission values occurred in the MDS regardless of whether they were from the random or local burning regimes. A key difference between local and random burning was that the former had higher fuel moisture levels; as such, we assume that the elevated methane emissions in the MDS were due to the higher green leaf content combusted during this season. We also found that the LDS fires had the lowest values of the $CH_4$ EF and ED even though the MCE values for the LDS were also the lowest. This suggests that dry leaf litter causes a drop in the MCE without a corresponding rise in the $CH_4$ emissions (indeed, they fell) due to the low fuel moisture and humidity levels of the LDS fires. The EFs for CO and $CH_4$ were not correlated and the ratio of the EF CO to $CH_4$ varied by season. In summary, we conclude that the key factors governing methane emissions have complex relationships and differ by season.

This study finds that including the MDS in the analysis (as opposed to the oft-used EDS/LDS binary) is critical because the MDS has unique weather and fuel conditions that lead to unique relationships between key factors affecting emissions parameters. The MDS has higher winds and lower temperatures than the EDS and LDS, and there are key ecological differences especially concerning the leaf state. Importantly, we found that the relationship between the key factor of fire intensity and EF $CH_4$ differed by season; during the MDS, the fire intensity correlated positively with the EF $CH_4$ but this relationship was negative during the other seasons. Together, these findings suggest that in the MDS, elevated wind speeds affect burning in such a way as to increase both the methane emissions and MCE. This study also found that increasing the fire intensity correlated negatively, although insignificantly, with the EF $CH_4$ for all the fire types, while our previous research found a positive trend [4]. It is noteworthy that when the outliers of very high-intensity fires were included, the analysis found that the EF $CH_4$ did increase with the fire intensity. As such, we conclude that further research is needed to explore the relationship between high-intensity head fires and methane emissions. It is important to remember that while high-intensity fires are not desired by local populations, they do occasionally occur and our analysis indicates that these fires release higher amounts of methane.

We also conclude that while it may be possible to shift burning practices to reduce methane emissions from West African savanna fires in theory, in reality emissions from fires are a complex phenomenon governed by numerous factors and it does not appear that simply shifting the season of burning will have the desired effects and would likely have numerous negative effects. Indeed, earlier burning, if even possible, would likely result in higher methane emissions due to the combustion of greener fuels and especially green leaves and would likely require dangerous head fires.

Finally, our fieldwork and observations suggest that the burning of green tree leaves is a key source of methane emissions from savannas, as was found by [43] for nearby Senegal. In addition, we recommend that the percentage of leaf fall is developed as an indicator of the division between the MDS and LDS for further studies because the leaf state is an important determinant of emissions. We also note that the commonly used $CH_4$ EF values are derived primarily from grass-dominated savanna fires, although it is clear from studies in other biomes that woody vegetation has higher EF values (e.g., [47]) and that the fraction of tree-leaf litter and coverage of different savanna types remain understudied [48]. As such, we suggest that studies of emissions use drone-mounted gas sensors, which could capture emissions from different vegetation formations and would avoid the shortcomings of both ground-level and airborne measurements.

**Supplementary Materials:** The following supporting information can be downloaded at: https://www.mdpi.com/article/10.3390/fire6020052/s1.

**Author Contributions:** P.L. was the principal investigator of the project, supervised all aspects of the research, and wrote the manuscript. M.K. was involved in the fieldwork and in the gathering, cleaning, and organizing of all data as well as commenting on manuscript. F.D. was the head of the field research team and an advisor on the field. R.J. and L.Y. were involved in data organization and analysis as well as discussing and commenting on manuscript. C.M.R. was responsible for the statistical analysis with coding assistance from Q.L. F.C. was the indigenous environmental advisor. All authors have read and agreed to the published version of the manuscript.

**Funding:** This research was funded by the National Science Foundation U.S.A. (Grant number 1313820).

**Data Availability Statement:** https://cla.csulb.edu/departments/geography/savannalabo/data/ (accessed on 1 January 2020).

**Acknowledgments:** The authors wish to thank Facourou Camara for his never-ending help in the field, Umu Kante for keeping us all happy and fed, and the people of Tabou and Faradiélé for supporting this research.

**Conflicts of Interest:** The authors declare no conflict of interest. The funders had no role in the design of the study; in the collection, analyses, or interpretation of data; in the writing of the manuscript, or in the decision to publish the results.

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
