# Peer review of "The Pyrogeography of Methane Emissions from Seasonal Mosaic Burning Regimes in a West African Landscape"

_fire, doi:10.3390/fire6020052_

Round 1

Reviewer 1 Report

This is an important manuscript; the amount of data collected and analyzed is impressive. But, overall, the manuscript is long. The Introduction seemed a long and meandering description of the problem and study objective. The description of the results seemed diffuse making the narrative difficult to follow. The associated tables and figures were difficult to read and provided little assistance in following the authors’ reasoning. Perhaps, the data and figures could be presented as supplementary material and the Results section include the pertinent graphs only. The Discussion also seemed long; the Discussion could make more of an impact by clarifying the results. The section in the Discussion on Policy Implications is so crucial to good science. In fact, there was enough material to comprise a second manuscript. The manuscript subject and research was good, but there was so much data presented, it was difficult to follow the data, analysis, and reasoning to understand the authors’ conclusions.

Below, are a few general comments: 

·       If using more than one equation, they should be separated from the text and numbered.

·       Be sure the elements of figures/tables are explained in the caption. For example, Figure 3 is eye-catching but the application to the associated text is not clear. What does the * refer to in Table 2?

·       Since so many abbreviations are used in the narrative, a list abbreviations and definitions would be helpful. The readers may not be as familiar with the data as are the authors.

Author Response

We attached a detailed response to all 3 reviews.

Reviewer 2 Report

The study entitled “The Pyrogeography of Methane Emissions from Seasonal-Mo-2 saic Burning Regimes in a West African Landscape” is of research interest and represents a contribution to guide national policies regarding shifting local practices of burning to decrease Methane emissions from fires in West African Landscapes.

In my opinion, after authors reviewing the manuscript, I think that it can be considered for publishing in the Fire Journal. I recommend this study with minor revisions.

There are some questions related to compliance with the rules of the sections that make up the structure of a research paper. Here I list two:

1.      According to the Journal “Instructions for authors” abstracts must have a maximum of 200 words and must be a single paragraph, which the authors clearly did not follow in their study. As it is, the abstract is like an extended abstract;

2.      The conclusion section must be shortened. Authors may do some condensation of repeated information, and there is a long paragraph of discussion (from line 854 to line 870). Conclusions shouldn´t have references. Lines 871 to 884 can go to the discussion or even to the Introduction.

Please remove any other references from the Conclusion section; lines 886 and 886 are repeated content and should be moved to the Discussion; The same for lines between 889 and 894.

My major comment is related with the inexistence of a statistical analysis of outliers. Confidence in the results can be increased if there is a more substantiated statistical assessment of outliers, instead of relying in the quartile rule of outliers; authors should assess the impact of removing outliers on the conclusions of this study. What if they weren’t removed? How would main results align with previous and other authors results? Analysis of outliers could for example include comparing studentized deleted residuals to a T critical value using the Bonferroni alpha correction.

Also, figures can be improved and some of them have poor quality

Minor comments:

-        In the introduction the authors list two objectives, the last with a less broad perspective. Please consider changing these to more specific objectives. Also, the authors could build a research question related with the conclusions they obtained with this study.

-        Quality of pictures need to be improved (for example between 8 and 13)

-        Line 215: the first time MCE is presented with no translation of the acronym; it only appears in Line 284

-        Line 258: the authors should explain why they didn’t also consider random burns in the late dry season?

-        Line 272: it is the first time that authors mention that they are using satellite data; this should be mentioned before

-        Line 276: Other potential problems exist such as undetected understorey fires; low intensity fires and the time lag between fire occurrence and satellite passage, where vegetation recovery may decrease substantially the vegetation reflectance signal

-        Figure 5 needs to be improved

-        Line 341: and why there was not a sampling in the LDS?

-        Line 343: what research? Authors must add this

-        Line 347. Authors need to specify what work are they mentioning

-        Line 426: Figure 0? Where is the figure? There is only one equation. This paragraph is not clear. It seems that some paste was done here by mistake.

-        Line 444: replace fire speed by fire spread

-        Line 450-451: why are these lines in bold?

-        Line 461: table 1 – use two decimal places only when necessary. Why certain variables have an increased precision (example fuel moisture or scorch height). In the case of the last was it measured with two decimals?

-        Line 495: there should be caution when removing outliers. Is there any interpretation for those observations related with their values? Were they measurement errors? If they are possible observations they should never be removed just to improve results.

-        Line 512: table 2- authors should describe acronyms in the table caption; in units capital letters is for names, thus kW/m

-        Line 489: sometimes authors use Methane EF others CH4 EF; please uniformize

-        Line 664: still regarding my previous comment, this change in the trend of the relationship must be further explored with a deeper understanding of what makes these observations being outliers and if they should be removed or not from analysis. I would appreciate if the authors could consider this in their statistical analysis. It would be desirable to have more confidence in these relationships that somehow are opposite to what was mentioned on line 663.

-        Line 767-773: very long sentence; please break it because as it is is very difficult to follow

-        Figure 5 needs quality improvement; authors should also consider to map the burn date as the three classes of analysis: EDS, MDS and LDS; also all figure can be improved. As maps they must have scale and orientation. In the figure caption add areas to “two study…” What data is used in this figure? What is the land cover data source? There is also one parenthesis missing

Author Response

(The authors gave the same response as above.)

Reviewer 3 Report

The manuscript reports worth experimental fire control data for methane emissions in a West African Landscape.  It is well-written, and the organization of the manuscript is clear.

There are some modifications to be addressed before progressing further.

1.              Figure 3. Theoretical competing determinants of methane emissions in a savanna landscape by season.

It is not suitable to use a Tai Chi symbol here. There is no clear explanation of the linkage and consideration of intellectual property rights.

2.              Sensitivity of the experiments can be discussed a bit.

3. Results from other experiments, if any, should be compared with this study.

Author Response

(The authors gave the same response as above.)
